# Comparing Clinicopathological and Immunohistochemical Features of Colorectal Carcinoma between Young and Old Age Groups

**DOI:** 10.3390/diagnostics14161743

**Published:** 2024-08-11

**Authors:** Heyam Awad, Sanad Elshebli, Khaled Hasan, Yousef Eid, Fatima Obeidat, Mohammad Alzyoud, Basheer Alakhras, Faris AlShammas

**Affiliations:** 1Department of Histopathology, Microbiology and Forensic Medicine, University of Jordan, Amman 11942, Jordan; scnadsh@gmail.com (S.E.); khaled_ahmad.1995@yahoo.com (K.H.); yousef.eid@hotmail.com (Y.E.); fatima.obeidat1971@outlook.com (F.O.); 2Department of Lab Medicine, Jordan University Hospital, Amman 11942, Jordan; moha.alzyoud@gmail.com (M.A.); basheerradi@yahoo.com (B.A.); farisalshammas@yahoo.com (F.A.)

**Keywords:** colorectal carcinoma, immunohistochemistry, Jordan, mismatch repair, PD-L1, Ki67, P53, BRAF

## Abstract

The incidence of colorectal carcinoma (CRC) is increasing among individuals younger than 50, and some studies suggest the presence of differences in CRC among old and young individuals regarding clinical and histopathological features. The aim of this study was to compare clinicopathological features, mismatch repair protein status, and expression of certain immunohistochemical stains between young and old groups. The study included 180 cases and found significant histological and immunohistochemical differences between the two groups. CRC in the young tends to be more right-sided and has a higher percentage of dMMR proteins, but less expression of p53 mutations. These features are commoner in Lynch syndrome, and more investigations to study the relationship between young-onset CRC and hereditary syndromes are needed. Young-onset CRC also tends to show higher expression of tumor cell PD-L1, which is an expected finding, as dMMR cases are more likely to be immunogenic. Two other significant differences are the higher percentage of mucinous carcinoma and the higher tumor grade in young-onset CRC. These two features suggest a more advanced disease with possibly worse outcomes; however, there is no difference in disease stage between the two age groups.

## 1. Introduction

Colorectal carcinoma (CRC) is the second-most common cause of cancer death and the third-most diagnosed cancer in the world [1]. Screening programs and changes in lifestyle have resulted in decreased incidence of CRC in the developed countries [2]. In the United States, the incidence of CRC has decreased by 35% since 1990, when screening programs began to be widely used [3]. However, the incidence of CRC is increasing among individuals who are less than 50 years of age [4,5,6,7,8]. This trend was reported in many countries, including the United States, Canada [1], New Zealand [9], Australia [10], Europe [11], Japan [12], and Egypt [13]. Moreover, CRC is now the third-leading cause of cancer mortality in the below-50 age group [14]. Due to this rise in CRC incidence and mortality, screening guidelines recommend starting screening at 45 years instead of the previously recommended 50 [15,16,17].

Risk factors for CRC include obesity, decreased physical activity, increased intake of processed meat, sugary foods, and smoking [1,18]. The rising trend in obesity among young adults may be a factor of increased CRC in young patients [19,20].

The role of inheritance in CRC among young individuals is still controversial. One study found that 22% of CRC in young patients were due to inherited cancer syndromes [21], whereas a study from Egypt found no significant difference in family history of CRC among young and old patients, and hence concluded that hereditary factors are unlikely to be the cause of increased incidence of CRC among young individuals [13].

Some studies suggest that there are differences in CRC among old and young individuals regarding histopathological features and survival rates. The literature is contradictory regarding the effect of age on survival. Some researchers found that young patients were more likely to present with advanced disease [4]. Other studies reported better survival of patients with CRC at an early stage [22], whereas others showed that young patients had similar relative survival to older ones [23]. Moreover, histological features and genetic mutations were reported to differ between young and old patients [21]. For example, some studies documented a high percentage of young cases presenting with advanced-stage disease [24,25,26,27,28]. Others reported a high incidence of mucinous carcinoma at a young age [3,25,29]. On the other hand, some studies found comparable histological features between young and older individuals with CRC [13,29].

In Jordan, CRC is the most common cancer in males and the second in females, and is the second cause of cancer mortality in both sexes. There are no studies from Jordan that have investigated the differences of CRC between young and old patients.

The aims of this retrospective cohort study were to (i) compare the clinicopathological features of CRC among young and old groups, (ii) compare the mismatch repair (MMR) protein status between these two groups, and (iii) compare expression of certain immunohistochemical stains between the two groups, including p53, beta catenin, BRAF, Ki67, and PD-L1.

We choose a cut-off point of 50 years to divide the patients into young and old groups, as this is the point used by the majority of published studies and because it is the age of CRC screening [30].

The importance of this study stems from the fact that it is the first study from our region to investigate the biology of CRC and its behavior in the two age groups. Understanding the biology and behavior of CRC in young patients can aid in screening and management decisions.

## 2. Materials and Methods

### 2.1. Study Design and Ethical Issues

This retrospective cohort study was conducted at Jordan University Hospital (JUH) in Amman/Jordan. The study was approved by the University of Jordan Research Deanship and the Institutional Review Board (IRB) at JUH.

Electronic databases at the histopathology department at JUH were searched for CRC cases during the study period from 1 January 2013 till 31 May 2023. All cases of CRC from the young group (patients 50 years or younger) were selected (90 cases). Then, the same number (90 cases) of patients older than 50 was selected to have two equal groups.

The main inclusion criterion in both groups was primary colonic or rectal adenocarcinoma cases, regardless of subtype, grade, or stage. To obtain enough tissue to perform immunohistochemical analysis, resection specimens were used. Exclusion criteria were (i) primary colorectal malignancy of any histological type other than adenocarcinoma, for example, lymphoma, GIST, and neuroendocrine carcinoma, and (ii) cases where paraffin blocks were unavailable.

Using computerized patient records and histopathology reports, demographic and histopathological data were collected for all cases included. Also, tumor site, type, grade, and the presence or absence of perineural and lymphovascular invasion were documented. According to the site, tumors were labeled as left colon or right colon tumors. The right colon extends from the cecum to the splenic flexure. The left colon extends from the splenic flexure to the rectum.

Hematoxylin and eosin (H&E) slides were reviewed to confirm the diagnosis of CRC, document the subtype, verify the grade, and select a representative tissue block for staining. The histological review was conducted using an Olympus BX53 microscope (Olympus, Tokyo, Japan), with a high-power field diameter of 0.55 mm, and field area of 0.238 mm^2^.

### 2.2. Tissue Microarray

The immunohistochemical stains were performed on tissue microarray blocks. To decrease costs, we selected half the cases from each age group to perform immunohistochemical analysis, i.e., 45 cases from each group. The tissue microarray blocks were prepared from the selected paraffin blocks to obtain 2 mm fragments from each block, using microarray rubber, array mold RETA-IHC World, (catalogue number IW-110), and a wax dispenser machine (MEDITE-TES99).

### 2.3. Immunohistochemical Analysis

Each tissue microarray block was stained with the following immunohistochemical stains: ki67, p53, MSH2, MSH6, MLH1, PMS2, BRAF, beta catenin, and PD-L1. All stains were conducted according to the manufacturer’s protocols. The tissue microarray blocks were formalin-fixed and paraffin-embedded, cut into 4 μm-thick slices using a microtome (Leica, RM2125RT, Wetzlar, Germany), and put on positively charged glass slides. These slides were put into a Ventana medical systems (Roche, Basel, Switzerland) BenchMark-XT IHC/ISH and Ultra Ventana staining module. This system is fully automated. Briefly, the slides were baked overnight at 50 °C to be deparaffinized, then the antigens were retrieved by heat induced epitope retrieval (HIER) in Tris-EDTA buffer of pH 7.8 at 95 °C for 44 min. This was followed by incubation with the primary antibody for 60 min. OmniMap anti-Rb HRP was then used to provide a clean background. Staining was performed by applying one drop of DAB CM and one drop H_2_O_2_ CM, which were incubated for 8 min. Hematoxylin was used for counterstaining, and bluing reagent was used for post-counterstaining. The slides were then cover-slipped in permanent mounting media. All these processes were performed by a machine. The Ventana system uses the Universal DAB Detection Kit (760-500/05269806001) and OptiView DAB IHC Detection Kit (760-700/06396500001) for detecting the antibodies. The details of this system can be found on the Roche website [31]. For each stain, a primary monoclonal antibody was used to detect the antigen. A list of the antibodies used for each stain is provided in Table 1. Positive and negative controls were used, with each run for quality assurance purposes. Table 1 details the primary antibodies used for each stain, as well as the type of positive control tissue and staining pattern considered when reporting each stain. For each stain, the percentage of tumor cells stained positively was recorded.

To determine mismatch repair protein (MMR) status, cases were considered deficient (dMMR) if there were loss of one or more of the four MMR protein stains, namely: MSH6, MSH2, MLH1, and PMS2. Cases where the four stains were expressed in tumor cell nuclei were considered proficient (pMMR) [32].

PD-L1 stains were evaluated in tumor cells and in tumor-infiltrating lymphocytes (TILs). Staining in tumor cells was considered positive in cases where >1% of the cells expressed complete or partial membranous staining, whereas in tumor TILs, membranous or cytoplasmic positivity in >1% of cells was considered positive [33].

Regarding BRAF, which is a poor prognostic indicator in CRC [34], cytoplasmic staining of any intensity was considered positive [34]. As for beta catenin, nuclear staining of any intensity was considered positive [35]. Β-catenin is an oncogene commonly mutated in CRC and it plays important roles in cell proliferation and differentiation [36].

Ki67 is a proliferative marker that stains mitotically active nuclei. Ki67 staining was divided into high expression (positivity in ≥50% of nuclei), and low expression (positivity in <50% of nuclei) [37]. Figure 1 shows examples of positive and negative stains of BRAF, PD-L1, and beta catenin.

P53 protein expression in immunohistochemistry correlates with mutations in the TP53 gene. Two types of mutations are recognized in TP53, namely, complete deletion (loss of function) and missense mutations resulting in accumulation of abnormal mutated p53 protein [38]. As such, p53 immunostaining is considered abnormal if the nuclear staining is 0% or near 100%, with the 0% correlating with complete loss of function (deletion of TP53), and the 100% nuclear staining correlating with accumulation of mutated protein (missense mutation) [38,39]. Anything in between is considered wild type (not mutated). Figure 2 shows examples of mutated and wild-type p53 stains.

### 2.4. Statistical Analysis

The data were collated on a Microsoft Excel sheet, version 16.12. Categorical data are presented as numbers and percentages. The mean, median, and standard deviation (SD) were calculated for continuous data. The chi-squared (X^2^) test was used to compare the variables between the two age groups and a significant *p* value was considered to be <0.05.

## 3. Results

### 3.1. Cases

A total of 180 cases were included: half of these were 50 years old or younger (the young group), and the other half were older than 50 (the old group). The age range was 17–84, the mean was 53.0, the median was 51, and the SD was 15.8. A total of 72 (40%) cases were females and 108 (60%) were males. Among the young group, 37 (41%) were females and 53 (59%) were males. Among the old group 35 (39%) were females and 45 (61%) were males.

### 3.2. Histopathological Features

Most of the tumors (60 cases, 66.7%) were adenocarcinomas, the other 30 (33.3%) were mucinous carcinomas. Regarding tumor site, 95 (52.8%) were right-sided and 85 (47.2%) were left-sided.

Comparing tumor types, sites and histopathological features between the two age groups are detailed in Table 2. In the young group, there were more right-sided tumors (61%) compared to 44.4% in the old group. This difference was statistically significant, *p* value = 0.025. There was also a statistically significant difference between the two groups regarding histological type, with more mucinous carcinomas in the young group (22.2% of the young group tumors being mucinous compared to 10% in the old group). Moreover, 33.3% of the young group’s tumors were of high grade (grade 3) compared to only 8.9% in the old group, *p* value = 0.0001. There was no statistically significant difference between the two groups regarding gender, lymphovascular and perineural invasion stage, lymph node involvement (N stage), or distant metastases (M stage).

### 3.3. Mismatch Repair Protein (MMR) Status

Of the overall selected sample, 25 (27.8%) cases showed dMMR proteins. There was a statistically significant difference in MMR status between the young and old groups, *p* = 0.00, while 44.4% of the young group cases were deficient in MMR proteins compared to 11.1% of the old group.

### 3.4. Immunohistochemical Analysis

Several immunohistochemical stains were compared between the two age groups. There was a statistically significant difference in PD-L1 expression in tumor cells, *p* = 0.04, with 31.1% of the young group expressing tumor cell positivity with PD-L1 compared to 13.3% of the old group. However, there was no significant difference in PD-L1 expression in TILs between the two groups, *p* = 0.67.

Regarding P53 expression, 60% of the young group showed wild-type patterns of expression compared to 20% of the old group. Among the young group, 18 cases showed abnormal expression, in 7 of which p53 was totally negative, and in 11 cases, 100% of tumor cells showed p53 positivity, indicating accumulation of a mutated p53 protein. In the old group, a total of 36 cases showed abnormal p53 expression, 20 of which exhibited total loss of p53 protein, whereas 19 cases exhibited expression in 100% of tumor cells (mutated protein accumulation). These differences in p53 expression were statistically significant: protein expression was abnormal (lost or mutated) in 40% of the young group cases compared to 80% in the old group. *p* = 0.00.

There was no statistically significant difference in immunohistochemical expression of BRAF, beta catenin, or Ki67 stains between the two groups.

Table 3 details the differences in the studied immunohistochemical stains between the two age groups.

## 4. Discussion

This retrospective study of 180 cases is the first from Jordan to compare the clinicopathological features and immunohistochemical staining expression of CRC cases between young and old groups. Compared to the old group, CRC in the young is more likely to be right-sided, has a higher percentage of mucinous carcinoma type, is of a higher grade, and has a higher percentage of dMMR proteins and of tumor cell PD-L1, but a lower rate of abnormal p53 expression.

Among our cohort, 61% of CRCs in the young group were right-sided compared to 44.4% in the old group, and this difference was statistically significant, *p* = 0.03. This result contradicts previous research findings of predominantly left-sided tumors in young individuals [3,25,29]. This discrepancy could be explained by the high proportion of dMMR cases among our sample: around 44% of young cases in our cohort showed dMMR proteins. Right-sided tumors are known to have more mutations in mismatch repair genes than left-sided ones [40]. The high percentage of dMMR among Jordanian patients regardless of age was documented in previous research [41,42]. In this sample, around 28% of the overall cases showed dMMR proteins, which is higher than the reported percentage of 15% in Western countries [43,44].

Moreover, there was a statistically significant difference in MMR status between the young and old groups, *p* = 0.00. Around 44% of the young group cases were deficient in MMR proteins compared to 11% of the old group. Similar results were reported by Ballester et al. [21] and Chang et al. [45]. The reason behind the high proportion of dMMR in young patients can be attributed to inherited factors; however, this needs further investigation. Lynch syndrome is an autosomal-dominant disorder caused by mutations in one of the mismatch repair (MMR) genes, which include MLH1, MSH2, MSH6, and PMS2. Loss of MMR function leads to microsatellite instability (MSI) in tumors, which results in carcinogenesis [46]. Unfortunately, there are no published data regarding the incidence of Lynch syndrome in Jordan, and a recent study highlighted the suboptimal identification of Lynch syndrome in several Middle Eastern countries, including Jordan [46].

Another significant difference between the two groups was the histological type, with more mucinous carcinomas in the young group, *p* = 0.01. This result is in line with previous research that documented a high percentage of mucinous tumors in young individuals [12,13,21,47]. Moreover, the percentage of young patients with high-grade tumors is more than that of the old group, and this difference is statistically significant. Similar results of higher-grade tumors in young patients were reported in previous research [21]. The latter two findings, namely, the higher percentage of mucinous and high-grade tumors in the young group, might be explained by the high proportion of dMMR protein status. Recent research on a large cohort of patients has documented an association between these two features and dMMR status [48].

There was no statistically significant difference between the two groups regarding gender, lymphovascular and perineural invasion, T stage, lymph node involvement, or distant metastases. These results are similar to those reported from a UK study, which found comparable histological features between young and older individuals with CRC [29]. However, there are studies that documented differences in histological features, with a high percentage of young cases presenting with advanced-stage disease [24,25,26,27,28].

Regarding immunohistochemical analysis, there was a statistically significant difference in PD-L1 expression in tumor cells, *p* = 0.04. However, there was no significant difference in PD-L1 expression in TILs between the two groups, *p* = 0.67. This result is in line with previous research that documented higher PD-L1 expression in dMMR cases [49,50]. dMMR CRCs are more likely to benefit from immunotherapy, and hence this treatment option can be used to improve outcome of young patients with CRC.

Another difference in immunohistochemical stain expression is documented in P53. In the old group, 80% of cases showed abnormal p53 expression in the form of total loss or accumulation of mutated p53 compared to 40% of the young group showing abnormal p53 expression, *p* = 0.00. P53 mutations are commoner in left-sided CRC and follow the chromosomal instability molecular pathway [40], where tumorigenesis starts with inactivation of the adenomatous polyposis coli (APC) gene, followed by activation of KRAS and then inactivation of p53 mutations [38,40]. As more tumors in the old group in our cohort were left-sided, this result of more p53 mutations in the old age was expected.

In our study There was no statistically significant difference in immunohistochemical expression of BRAF, beta catenin, or Ki67 stains between the two groups.

Our results suggest that CRC in young age follows the microsatellite instability pathway, supported by the high percentage of dMMR and the right-sidedness, whereas CRC in old age follows the chromosomal instability pathway, as supported by high p53 mutation rate and the left-sidedness of the tumors. This result needs more investigation at the genetic level to determine its validity. The role of inheritance also needs more exploration. Right-sided tumors with microsatellite instability occur in Lynch syndrome, but we do not have full data regarding family history of CRC among our sample to peruse this further. The other differences of high proportion of mucinous tumors and high-grade tumors are also associated with dMMR cases. Moreover, the higher PD-L1 expression in the young group could also be related to the high proportion of dMMR cases. As such, it seems that all the differences between the two age groups are related to the MMR status. This result needs more investigation to determine its validity.

The importance of this study stems from the fact that it is the first study from our region to investigate the biology of CRC and its behavior in these two age groups. Understanding the biology and behavior of CRC in young patients can aid in screening and management decisions.

This study has some limitations, including being a single-institution study with a relatively small sample. Another limitation is the lack of data regarding family history. Further research addressing the differences in CRC between young and old individuals at the genetic level is needed to investigate the findings from this work and previous published research. Moreover, more research on the role of inheritance in the development of CRC among young individuals is needed.

## 5. Conclusions

CRC in young individuals has some histological and immunohistochemical differences from older ones, and they might have different mutagenic pathways. Young-onset CRC tends to be more right-sided and has a higher percentage of dMMR proteins, but less expression of p53 mutations. These three features are commoner in Lynch syndrome, and more investigations to study the relationship between young onset CRC and hereditary syndromes are needed. Young-onset CRC also tends to have higher expression of PD-L1 tumor cell, which is an expected finding, as dMMR cases are more likely to be immunogenic and hence have a better response to immunotherapy. Two other significant differences between old and young patients are the higher percentage of mucinous carcinoma and the higher grade in young patients. Again, these two features were found to be associated with dMMR CRC cases.

## Figures and Tables

**Figure 1 diagnostics-14-01743-f001:**
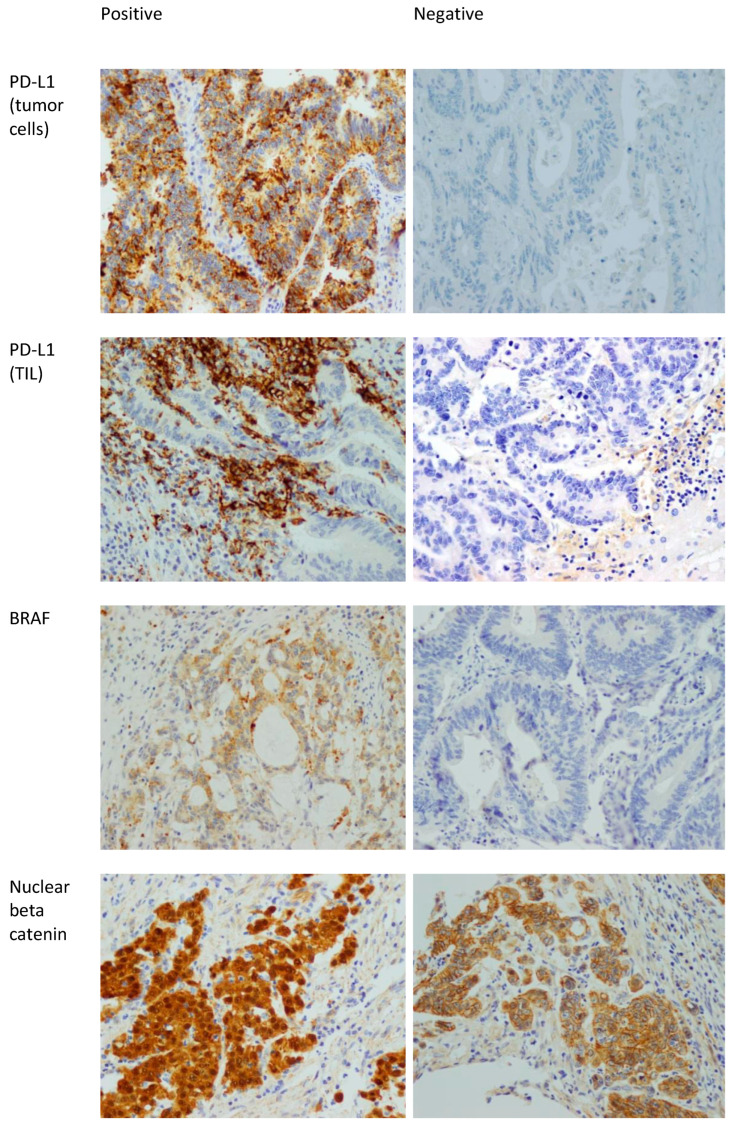
Positive and negative examples of BRAF, PD-L1 in tumor cells, PD-L1 in TILs, and beta catenin stains. 400× magnification. The positive staining in the beta catenin is nuclear. The presence of membranous staining (which is seen in the beta catenin picture labeled as negative) is normal in colonic mucosa and CRC. Note that the nuclei in that picture are negative.

**Figure 2 diagnostics-14-01743-f002:**
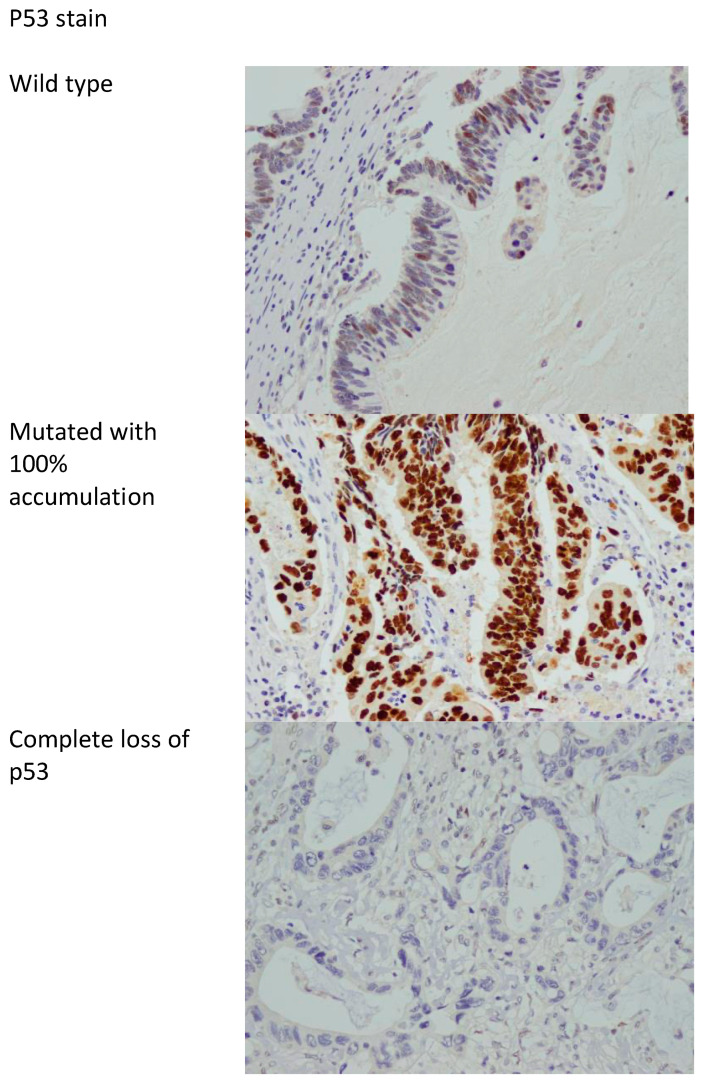
p53 stain showing the wild type (non-mutated) and mutated with accumulation of abnormal p53 protein in 100% of cells and complete loss of p53 stain. 400× magnification.

**Table 1 diagnostics-14-01743-t001:** The antibodies used for immunohistochemical stains and the type of positive control tissue and staining pattern considered when reporting each stain.

Stain	Antibody Used	Positive Control	Staining Pattern
MSH 6	Clone 44 (Roche) (Basel, Switzerland)	Colon	Nuclear
MLH 1	Clone M1 (Roche)	Colon	Nuclear
MSH 2	Clone G219-1129 (Roche)	Colon	Nuclear
PMS 2	Clone EPR3947 (Roche)	Colon	Nuclear
PD-L1	Clone 22C.3 (Dako) (Glostrup, Denmark)	Placenta and tonsil	Membranous in tumor cells. Membranous or cytoplasmic in TILs
BRAF	BRAF V600E (VE1) (Roche)	Melanoma	Cytoplasmic
Ki67	Clone 30-9 (Roche)	Lymphoma	Nuclear
P53	Clone DO7, (Roche)	Ovarian tumor	Nuclear
Beta catenin	Clone 14 (Sigma-Aldrich) (Burlington, MA, USA)	Colonic cancer	Nuclear

**Table 2 diagnostics-14-01743-t002:** Comparison of tumor types, sites, and histopathological features between the two age groups.

	Young Group Total: 90	Old Group Total: 90	
Characteristic	Number (%)	Number (%)	*p* Value
Gender			0.73
Female	37 (41)	35 (39)	
Male	53 (59)	45 (61)	
Site			0.03
Rt colon	55 (61)	40 (44.4)	
Lt colon	35 (38.9)	50 (55.6)	
Histological type			0.01
Adenocarcinoma	70 (77.8)	81 (90)	
Mucinous	20 (22.2)	9 (10)	
Grade			0.00
Low grade	60 (66.7)	82 (91.1)	
High grade	30 (33.3)	8 (8.9)	
LVA			0.76
Present	31	33	
Absent	59	57	
Perineural invasion			0.28
Present	75 (83.3)	80 (88.9)	
Absent	15 (16.7)	10 (11.1)	
T stage			0.11
T1/T2	11 (12.2)	19 (21.1)	
T3/T4	79 (87.8)	71 (78.9)	
N stage			0.88
N0	41	40 (44.4)	
N1/N2	49	50 (55.6)	
M stage *			0.24
M1	10 (33.3)	17 (29.8)	
M2	20 (66.7)	40 (70.2)	

* M stage was documented in 87 cases; these were included in the statistics regarding metastases. For the rest of the cases, M stage was not known.

**Table 3 diagnostics-14-01743-t003:** Comparison of immunohistochemical stains between the two age groups.

	Young Group Total: 45	Old Group Total: 45	
Stain	Number (%)	Number (%)	*p* Value
dMMR	20 (44.4)	5 (11.1)	0.00
pMMR	25 (55.6)	40 (88.9)	
PD-L1 in tumor cells			0.04
Positive	14 (31.1)	6 (13.3)	
negative	31 (68.9)	39 (86.7)	
PD-L1 in TILs			0.67
Positive	23 (51.1)	21 (46.7)	
Negative	22 (48.9)	24 (53.3)	
BRAF			1.00
Positive	7 (15.6)	7 (15.6)	
Negative	38 (84.4)	38 (84.4)	
Nuclear beta catenin			0.15
Positive	9 (20)	15 (33.3)	
Negative	36 (80)	30 (66.7)	
P53			0.00
Lost or mutated	18 (40)	36 (80)	
Wild type	27 (60)	9 (20)	
Ki67			0.83
High >50%	23 (51.1)	22 (48.9)	
Low	22 (48.9)	23 (51.1)	

## Data Availability

Data of this research are available upon request from the corresponding author.

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
