# Peer review of "Comparing Clinicopathological and Immunohistochemical Features of Colorectal Carcinoma between Young and Old Age Groups"

_diagnostics, 2024, doi:10.3390/diagnostics14161743_

Round 1

Reviewer 1 Report

Comments and Suggestions for Authors

This study is interesting and shows the characterization of a colorectal cancer cohort in Jordan, which can contribute to the overall picture of the incidence and prevalence of colorectal cancer worldwide. The authors were unable to prove their hypothesis of a significant difference between young and older patients with colorectal cancer, which might have been helpful in stratifying their treatment strategies. Neverhteless, the demonstration od negative result is brave and should be encouraged.  The most dramatic flaw in this paper is the figures. They were created in an inappropriate application (possibly Word) as they contain some signs of world corrections (red wavy line). In addition, the photomicrographs are different sizes and have different margins, which make them confusing. Also, some microphotos have been manually corrected (stretched), which should be prohibited. Furthermore, the authors used the same magnification marks for all images, although it could be noticed that the magnifications are different because the size of the glands is very different. In Figure 2, the authors have shown a positive and negative expression of beta'catenin while the two images of beta-catenin actually show a positive expression. The whole confusion should therefore be corrected.

Author Response

Reviewer 1

Comment:

This study is interesting and shows the characterization of a colorectal cancer cohort in Jordan, which can contribute to the overall picture of the incidence and prevalence of colorectal cancer worldwide. The authors were unable to prove their hypothesis of a significant difference between young and older patients with colorectal cancer, which might have been helpful in stratifying their treatment strategies. Neverhteless, the demonstration of negative result is brave and should be encouraged.  The most dramatic flaw in this paper is the figures. They were created in an inappropriate application (possibly Word) as they contain some signs of world corrections (red wavy line). In addition, the photomicrographs are different sizes and have different margins, which make them confusing. Also, some microphotos have been manually corrected (stretched), which should be prohibited. Furthermore, the authors used the same magnification marks for all images, although it could be noticed that the magnifications are different because the size of the glands is very different. In Figure 2, the authors have shown a positive and negative expression of beta'catenin while the two images of beta-catenin actually show a positive expression. The whole confusion should therefore be corrected.

Response:

We thank the reviewer for carefully assessing our work. The comment on encouraging negative results is highly appreciated.

Regarding the figures, we totally agree with the reviewer, and we have made the corrections the reviewer asked for. All images are now in a picture format, and we made sure to take new pictures, all with 400X magnification.

Regarding beta catenin, it is normally present in the cell membrane of colonic cells, and this is not considered mutated. We replaced the picture with another one which is clearer regarding the negative staining in the nuclei. We added a comment regarding this so readers will not be confused and we made it clear that positivity of beta catenin refers to nuclear staining not membranous.

Reviewer 2 Report

Comments and Suggestions for Authors

The authors analyzed association of clinicopathological parameters and immunohistochemical markers of CRC in relation to the patients age in Jordanian population. The study provides interesting results. However, the novelty of the study is questionable and the results/conclusion could be biased by relatively small number of patients included in the study. The conclusion could be supported by the results but the discussion must be more precise and critical in relation to the presented findings. 

Discussion must underlie the novelty od this study and future directions.
Study shall be also analyzed regarding its weak points and these issues must be discussed. Detailed description of results including values, p-values or percentage are not recommended in the Discussion. 

What is the incidence of Lynch syndrome in colorectal patients in Jordan in comparison to the other countries? Please compare the data and, if relevant, include in the discussion.

Materials and methods must be carefully revised. The methods for immunostaining must be described in details in order to allow for their reproduction by another researchers. Provide more data about primary and secondary antibodies used such as dilutions, time of incubation, epitope retrieval method used etc.

Authors can consider citation/discussion of another relevant article that was recently published: https://doi.org/10.1007/s00384-024-04674-z

Minor remarks:

Unify the layout of figures/microphotographs. They don't look professional in the current form.
Names of genes/protein must be unified since they differ across the MS.
Correct in-text citations format (brackets before the full stop sign).

Comments on the Quality of English Language

The MS needs minor correction of English. I also recommend to replace "immunohistochemical stains" by another words (analysis, reactions etc) since the IHC reaction is not equivalent to simple staining procedure.

Author Response

Reviewer 2:

Comment 1:

The authors analyzed association of clinicopathological parameters and immunohistochemical markers of CRC in relation to the patients age in Jordanian population. The study provides interesting results. However, the novelty of the study is questionable and the results/conclusion could be biased by relatively small number of patients included in the study. The conclusion could be supported by the results but the discussion must be more precise and critical in relation to the presented findings. 

Response 1: We thank the reviewer for taking the time to carefully assess our manuscript. We agree that the discussion can be improved, and we have revised it to meet the reviewer’s request.

Comment 2:

Discussion must underlie the novelty of this study and future directions.
Study shall be also analyzed regarding its weak points and these issues must be discussed. Detailed description of results including values, p-values or percentage are not recommended in the Discussion. 

Response 2: Again, we agree about these comments regarding the discussion, so we added a description of the novelty of the study and suggested future directions. The limitations were also added and as requested detailed percentages and p values were omitted.

Comment 3:

What is the incidence of Lynch syndrome in colorectal patients in Jordan in comparison to the other countries? Please compare the data and, if relevant, include in the discussion.

Response 3: Unfortunately, there are no available data about the incidence of Lynch syndrome in Jordan. However, we added a paragraph about Lynch syndrome and discussed the lack of statistics about it and linked our findings to features seen in Lynch syndrome. We also suggested further research on this topic to investigate the role of inheritance in CRC among young individuals.

Comment 4:

Materials and methods must be carefully revised. The methods for immunostaining must be described in details in order to allow for their reproduction by another researchers. Provide more data about primary and secondary antibodies used such as dilutions, time of incubation, epitope retrieval method used etc.

Response 4: The immunostaining was fully automated. The technicians cut the tissue, put it on a slide and then the rest of the work is performed by the machine. We now made this clear in the methods sections. The details of the primary antibodies are present within table 1. We added a reference (the manufacturer website) for anyone who likes to check howthe system works.

Comment 5:

Authors can consider citation/discussion of another relevant article that was recently published: https://doi.org/10.1007/s00384-024-04674-z

Response 5:

We thank the reviewer for this suggestion. The article is very useful and was included in our citations.

Coment 6:

Minor remarks:

Unify the layout of figures/microphotographs. They don't look professional in the current form.
Names of genes/protein must be unified since they differ across the MS.
Correct in-text citations format (brackets before the full stop sign).

Response 6: All the requested corrections are done.

Comment 7:

Comments on the Quality of English Language. The MS needs minor correction of English. I also recommend replacing "immunohistochemical stains" by another words (analysis, reactions etc) since the IHC reaction is not equivalent to simple staining procedure.

Response 7: We have revised the language of the manuscript and we corrected the immunohistochemical stain and used analysis instead.

Round 2

Reviewer 2 Report

Comments and Suggestions for Authors

The authors correctly addressed most of reviewer's remarks. However, in reviwer's opinion the methodology of immunostaining needs additional revision. Brief description of the procedure used, unless the process is partially automated, is obligatory. The study is based on immunohistochemistry and without clear description of the method i cannot be accepted.

Minor remarks:

Table 1, 2nd column: "clone" or "Clone" - use capital letters consequently.

Symbols of proteins/genes: in the Table 1 there are spaces and additional characters while they are not present in the text or images. For example: table 1: “PD-L 1” while in the text above (line 115) PD-L1. In Table 3, 1st column: “PD-L 1 in tumor cells” while below, the same column “PD-L1” (without space). Please revise the whole MS including tables, figures and their descriptions and unify those names/symbols.

In figure 1 and 2 names of target proteins and staining result (positive/negative) is given next to the image, while in figure 3 images are marked by letters and the staining results is described below, in the legend. Consider using the same style of denoting for all images.

Whenever the exact p value is given, in the text or in tables/figures, it is recommended to provide each value with the same number of decimal places; e.g. tables 2 and 3.

Minor remarks:

Table 1, 2nd column: "clone" or "Clone" - use capital letters consiquently 

Author Response

Comment 1: The authors correctly addressed most of reviewer's remarks. However, in reviwer's opinion the methodology of immunostaining needs additional revision. Brief description of the procedure used, unless the process is partially automated, is obligatory. The study is based on immunohistochemistry and without clear description of the method i cannot be accepted.

Response 1:

We thank the reviewer for taking the time to review the manuscript again. We agree that description of the immunostaining procedure is essential. Our staining is fully automated; we cut the tissue and put it on positively charged slides then we feed these into the Ventana system where all the process is automated. We use this method in our lab to diagnose patients’ slides and for research purposes. We added a brief description of the procedure and explained that the immunohistochemical antigen retrieval, deparaffination and staining are all automated. We also added details of the detection system used by this method. We hope this is satisfactory.

Minor remarks:

Comment 2: Table 1, 2nd column: "clone" or "Clone" - use capital letters consequently.

Response 2: Corrected as requested.

Comment 3: Symbols of proteins/genes: in the Table 1 there are spaces and additional characters while they are not present in the text or images. For example: table 1: “PD-L 1” while in the text above (line 115) PD-L1. In Table 3, 1st column: “PD-L 1 in tumor cells” while below, the same column “PD-L1” (without space). Please revise the whole MS including tables, figures and their descriptions and unify those names/symbols.

Response 3: We thank the reviewer for this comment. We reviewed the whole manuscript and made the requested changes.

Comment 4: In figure 1 and 2 names of target proteins and staining result (positive/negative) is given next to the image, while in figure 3 images are marked by letters and the staining results is described below, in the legend. Consider using the same style of denoting for all images.

Response 4: We thank the reviewer for this important observation. We changed the layout of figure 3 to be in line with figures 1 and 2.

Comment 5: Whenever the exact p value is given, in the text or in tables/figures, it is recommended to provide each value with the same number of decimal places, e.g. tables 2 and 3.

Response 5: Done as requested. All p values are now given to 2 decimal points.

Minor remarks:

Table 1, 2nd column: "clone" or "Clone" - use capital letters consequently

Response: Done as requested. 
